# Pediatric Hyperglycemic Hyperosmolar Syndrome: A Comprehensive Approach to Diagnosis, Management, and Complications Utilizing Novel Summarizing Acronyms

**DOI:** 10.3390/children10111773

**Published:** 2023-10-31

**Authors:** Naser Amin Zahran, Shaheen Jadidi

**Affiliations:** 1Lurie Children’s at Northwestern Medicine Central DuPage Hospital, Winfield, IL 60190, USA; nzahran@luriechildrens.org; 2Loyola University Medical Center, Maywood, IL 60153, USA

**Keywords:** hyperglycemic hyperosmolar state (HHS), diabetic ketoacidosis (DKA), diabetes mellitus, hyperglycemia, mixed HHS and DKA, treatment protocols, prognosis, electrolyte imbalances, patient management strategies, dehydration

## Abstract

This paper focuses on hyperglycemic hyperosmolar syndrome (HHS), a unique hyperglycemic state requiring divergent diagnosis and treatment approaches from diabetic ketoacidosis (DKA) despite some shared characteristics. We introduce the mnemonic DI-FF-ER-EN-CE-S to encapsulate unique HHS management and complications. ‘DI’ emphasizes the need to delay and decrease initial insulin therapy until serum glucose decline is managed by fluid resuscitation alone. ‘FF’ stresses the importance of double fluid replacement compared to DKA due to severe dehydration and ‘ER’ electrolyte replacement due to profound losses and imbalances. ‘EN’ denotes the potential for encephalopathy and the requirement for a controlled serum osmolality reduction. ‘CE’ indicates cerebral edema, a rare complication in HHS. ‘S’ signifies systemic multiorgan failure. We categorize the associated risks into three mnemonic groups: the 3Rs (renal failure, respiratory distress, rhabdomyolysis), the 3Hs (heart failure, hypercoagulation, hyperthermia), and AP (arrhythmias, pancreatitis) to facilitate awareness and screening of HHS.

## 1. Introduction

Asymptomatic hyperglycemia is a phenomenon frequently observed in biochemistry profiles, typically presenting as mild and transient, with minimal clinical implications. Nevertheless, a wide range of triggers and etiologies can contribute to elevated blood glucose levels, including serious medical conditions and endocrine disorders, necessitating thorough evaluation and timely intervention [1]. To optimize the screening process and reduce the underdiagnosis of infrequent causes and ambiguous cases, we present a classification of multiple etiologies and various hyperglycemic states in Figure 1 (Algorithm). HHS and DKA are critical hyperglycemic emergencies. They are distinct conditions but are often mistaken for each other [2]. Historically, HHS has been primarily associated with adult patients with type 2 diabetes; however, recent studies have reported an emerging trend in pediatric populations, including those with type 1 diabetes [3].

The prevalence of HHS is found to be less common than DKA, ranging from 0.8 to 2 cases per 1000 hospital admissions, and typically affects older populations. On the other hand, DKA occurs more frequently, especially among type 1 diabetics, contributing to significant morbidity and mortality [4]. The current state of research has been focusing on differentiating between these two conditions, understanding their pathophysiology, and developing optimal management strategies. Delayed recognition and treatment of HHS can exacerbate risks and negatively impact patient outcomes. HHS can present across a spectrum, including cases that exhibit characteristics of both HHS and DKA [5]. The similarity in presentation between HHS and DKA often leads to confusion, resulting in potential mismanagement.

Increasing awareness of HHS and incorporating it into the differential diagnosis of severe hyperglycemia is crucial for several reasons. Early identification and intervention can significantly improve patient outcomes and reduce healthcare costs. Moreover, distinguishing between HHS and DKA allows for more targeted treatment and better utilization of medical resources. Understanding these conditions contributes to a broader recognition of the various triggers and etiologies of hyperglycemia, enhancing overall patient care and prevention strategies.

## 2. Review of HHS and DKA

Various terminologies have been employed historically in the medical literature to describe HHS, ranging from the broadly implicative “hyperosmolar hyperglycemic crisis” to more limiting and partially inclusive terms such as “hyperglycemic nonketotic syndrome or state”, “hyperosmolar nonacidotic uncontrolled diabetes”, and “hyperglycemic hyperosmolar coma” [6]. These appellations represent aspects of the intricate hyperglycemic state, which may manifest in either new-onset or known diabetes mellitus and typically evolve acutely or indolently, potentially culminating in life-threatening emergencies [4]. In comparison to DKA, HHS and mixed hyperosmolar diabetic ketoacidosis are less prevalent, though they carry a higher mortality rate and exemplify the most severe hyperglycemic emergencies [5]. While substantial differences and predominantly unique characteristics exist between DKA and HHS, some cases may exhibit overlapping pathogenesis, manifestations, and complications [7]. Despite the increasing incidence of HHS, it is often misdiagnosed initially as DKA due to inadequate awareness about HHS and its exclusion from the differential diagnosis of complicated diabetes [2]. This oversight precludes early recognition, timely monitoring, risk anticipation, and preventive therapy.

Classical HHS typically evolves gradually and follows a more protracted course than DKA [2]. HHS components are profound dehydration and electrolyte losses, extreme hyperglycemia, hyperosmolality, and altered mental status in the absence of significant ketosis and corresponding acidosis [2]. Table 1 delineates the Pediatric Endocrine Society’s diagnostic criteria for distinguishing HHS from other hyperglycemia forms. While DKA is associated with serum glucose levels exceeding 200 mg/dL, bicarbonate levels are usually below 15 mmol/L, and venous pH falls below 7.3. Additionally, DKA is characterized by the presence of ketonemia, ketonuria, and glucosuria [8].

Historically, HHS has been more commonly documented in elderly, obese, African American, and type 2 diabetic patients [9]. However, recent literature suggests an increasing incidence of HHS in younger, non-obese, and insulin-dependent diabetic patients. Male predominance, a poor social health network, and a positive family history of type 2 diabetes have been reported as risk factors [10]. The ingestion of large volumes of sugar-rich fluids, patients on diabetogenic medications including long-term systemic steroids, L-asparaginase, tacrolimus, growth hormone, and atypical antipsychotics, in addition to cystic fibrosis, have been identified as tangible triggers of HHS in some patients [11,12]. An increased risk with rare diseases, such as Kearns-Sayre syndrome, has also been published [13]. In many cases, concurrent infection and a lack of compliance are the commonly known precipitating triggers [4].

Any physician managing pediatric patients can encounter HHS; though it is uncommon, it is not rare. In one report, 2–4% of pediatric patients with type 2 diabetes had HHS at diagnosis [14]. Despite the sparse statistical data, substantially rising numbers of HHS in patients with type 1 diabetes continue to be published [15]. HHS admissions in some reports are increasing by around 4% annually [16].

Both HHS and DKA are due to insulin deficiency. The insulin level in HHS is low enough to induce extracellular hyperglycemia by decreasing tissue utilization and stimulating the release of glucagon and other counter-regulatory stress hormones to initiate glycogenolysis and gluconeogenesis (like DKA). However, in contrast to DKA, the insulin drop in HHS remains above the critical threshold of ketone formation through catabolic lipolysis and fatty acid oxidation. Hypothetically, the indolent HHS state is characterized by a prolonged, intermediately low insulin level that fails to meet glucose utilization demands, though the insulin nadir is not critically low and the subsequent cellular starvation from glucose deprivation is not sufficient to trigger compensatory adipose tissue oxidation and liver ketone production as an alternative energy source [17].

In DKA, the catabolic mechanism leads to an elevation in ketone bodies composed of acetone, acetoacetic acid, and beta-3-hydroxybutyrate, whereas HHS lacks significant ketoacidosis. The relatively longer period of symptomatic hyperglycemia in HHS produces an extracellular osmotic gradient, shifting fluids out of the cells and precipitating osmotic diuresis; coupled with poor fluid intake, there is an exacerbation of the negative fluid balance, resulting in extreme dehydration and severe hyperosmolar hyperglycemia without significant ketoacidosis [10]. This represents the main distinguishing difference between HHS and DKA. Longstanding and profound osmotic diuresis in HHS ultimately diminishes renal function, accelerating electrolyte waste to a greater extent than seen in DKA. This leads to significant sodium, chloride, potassium, phosphate, and magnesium tissue depletion that is not necessarily reflected in laboratory intravascular level measurements at the time of severe volume contraction [17]. Some of these derangements, particularly hypophosphatemia, have been implicated in Na-K-ATPase imbalance, which contributes to rhabdomyolysis in some HHS patients. Hyperglycemia itself activates a systemic stress response characterized by rising levels of inflammatory markers, including interleukins and tumor necrosis factor, among others [4,18]. These inflammatory markers further exacerbate the clinical manifestations of HHS and contribute to its morbidity and mortality.

## 3. The Challenge of Differentiating HHS and DKA at Clinical Presentation

Formulating a diagnosis and distinguishing between HHS and DKA purely based on clinical symptoms is an intricate undertaking, in spite of identifiable patterns in their progression, frequency, and associated risk factors. There is a frequent delay in the diagnosis of HHS, with patients often being initially treated as though they have DKA [2]. DKA commonly manifests in a much quicker timeframe, evolving over mere hours or days, whereas HHS generally pursues a more gradual course, unfolding over several days to weeks, notably in instances of type 2 diabetes [19]. Both conditions share common symptoms, such as polyuria, polydipsia, weight loss, and fatigue. However, DKA is often associated with vomiting, widespread abdominal discomfort, and, in severe cases, hyperventilation. Conversely, HHS may present itself through varying degrees of cognitive impairment and even seizures. Encephalopathy is generally seen when serum osmolality surpasses 330 mOsmol/kg and sodium levels climb over 160 mEq/L [4]. It has been reported that less than one-third of patients with HHS enter a comatose state [20]. Indications of dehydration in HHS may be camouflaged by hyperosmolality, potentially appearing less evident clinically and being underappreciated in overweight patients, despite their suffering from intense fluid losses that exceed those observed in DKA dehydration [21]. For patients demonstrating mild to severe symptoms of either condition, the associated signs could merge, creating a unique amalgamation of varying characteristics, similarities, and severity levels (Figure 2). This presents a fascinating mosaic portrait of each patient’s unique disease presentation.

Hyperglycemia can lead to dilutional hyponatremia, and the actual corrected serum sodium is calculated by adding 1.6 mEq/L for each 100 mg/dL glucose elevation above the normal serum glucose of 100 mg/dL [22]. Head imaging for suspected cerebral edema is rarely positive despite severe hyperglycemia, and brain swelling is less common in HHS than in DKA, even during treatment. This distinction, though rare, is vital as it emphasizes the fundamental differences in pathophysiology between HHS and DKA. If it occurs, brain edema can have severe consequences, necessitating prompt recognition and immediate appropriate intervention [23]. This intervention can be a distinguishing feature in treatment approaches for HHS and DKA, as early identification might lead to specific management strategies.

Muir et al. previously described abnormal verbal or motor response to pain, posturing, abnormal pupillary response, cranial nerve deficits (especially the III, IV, and VI nerves), abnormal respiratory patterns, and decreased oxygen saturation as signs of brain edema. They proposed a list of alarming signs and criteria for the clinical diagnosis of brain edema, with the major criteria including (1) altered mental status; (2) a sustained heart rate decrease of more than 20 bpm unexplained by other factors; and (3) age-inappropriate incontinence (4). The minor criteria encompass (1) ensuing vomiting or worsening severity if present before initial therapy despite treatment; (2) worsening headache; (3) lethargy; (4) increasing blood pressure with diastolic blood pressure (DBP) >90 mmHg; and (5) younger age (<5 years) [24]. These criteria can be utilized to avoid missing imminent cerebral edema, based on the presence of two major criteria or one major and two minor criteria.

The detailed explanation of the diagnosis criteria of brain edema is essential as it sheds light on the different manifestations and complexities that can be encountered in both HHS and DKA. Although brain edema is less commonly associated with HHS, the signs and criteria described are vital in distinguishing between these two conditions, enhancing clinical understanding and management. Recognizing these signs can guide clinicians in tailoring the therapeutic approach for each condition, ultimately contributing to improved patient outcomes and the reduction of adverse effects associated with misdiagnosis or delayed intervention. By delineating these specific differences, this information strengthens the foundation for more nuanced and effective care in the multifaceted landscape of hyperglycemic emergencies.

## 4. Complications in HHS and DKA

Cerebral edema is a severe complication associated with DKA and is observed in about 1% of patients, though there may be more subclinical instances. Its occurrence is less common in HHS [25]. This condition tends to emerge within the first 24 h of DKA treatment, with a peak incidence between 3 and 12 h [26]. While earlier beliefs linked its development to rapid rehydration or osmolality changes from insulin therapy, newer research offers different insights. Animal studies on hyperglycemia treatment revealed intracerebral accumulations of cations and complex carbohydrates. These could disrupt neuron membrane sodium–proton ion exchange, thus intensifying brain edema during fluid administration [27,28]. However, this theory does not explain why some patients exhibit cerebral edema before any treatment. Other research points to factors like hyperosmolality, cerebral hypoperfusion, antidiuretic hormone imbalance, and hyponatremia as initiators of brain edema [29]. There are also suggestions that the brain’s natural osmole defenses become overloaded, leading to imbalances in cerebral fluid levels [30]. Rapid changes in plasma osmolality can disturb cerebral euvolemia, creating an osmotic gradient that causes intracerebral fluid shifts and resultant brain edema [31]. Furthermore, some studies suggest that the extent of hypocapnia might be a predictor for brain edema onset [32].

Rhabdomyolysis is a rare condition, encountered more frequently in HHS than in DKA. Statistically, it correlates with elevated blood urea nitrogen (BUN) and creatinine levels as well as severe hyperglycemia and hypophosphatemia [33]. In extreme cases, it can potentially lead to a high risk of compartment syndrome and renal failure [34]. Disturbances in electrolytes and hypokalemic arrhythmias have been reported [35]. Malignant hyperthermia is a rare phenomenon characterized by a temperature exceeding 40 °C due to muscle calcium flux disruption, causing a significant shift from the sarcoplasmic reticulum to the myoplasm16. Several pediatric HHS cases complicated by hyperthermia have been documented, with a high mortality rate [36]. Various etiologies have been proposed, including rhabdomyolysis; however, the lack of muscle breakdown in some hyperthermic patients challenges this theory. In the past, insulin cresol preservative was implicated as a potential cause, supported by animal studies demonstrating hyperthermia induction [37]. Nevertheless, in certain cases, hyperthermia occurred prior to insulin therapy initiation, suggesting an inflammatory effect of cytokines and the theoretical possibility of central thermoregulation impairment.

Cardiogenic shock and pancreatitis are more severe and frequent complications in new-onset type 2 diabetes with HHS and occur rarely in pediatric type 1 DKA [38]. The risk of thrombosis is heightened in both DKA and HHS due to the hyperosmolar dehydration state. In some cases, thrombosis has revealed previously unknown underlying thrombophilia, particularly factor V Leiden gene mutation; in these instances, central line placement may increase the risk of thrombosis [39]. Respiratory distress syndrome and seizures, although rare, may present as complications in both DKA and HHS [40]. It is important for clinicians to recognize and promptly address these complications to improve patient outcomes.

Additionally, it is crucial to consider various risk factors and comorbidities when managing patients with DKA and HHS. For instance, age, sex, ethnicity, and the presence of underlying medical conditions may influence the development and progression of these hyperglycemic emergencies [41]. A prompt and accurate diagnosis, followed by the appropriate therapeutic interventions, is essential to minimize the risk of complications and mortality.

## 5. Treatment of HHS

The early recognition and timely diagnosis of HHS are the cornerstones of optimal therapy to achieve good outcomes and reduce complications. The treatment of HHS has evolved from DKA protocols, and there are no established standards for HHS therapy [40]. However, the 2022 ISPAD treatment recommendations for DKA, HHS, and mixed cases provide valuable guidance for managing these conditions [42].

Typical fluid loss in HHS averages 1.5 to 2 times that of DKA, making vigorous fluid management and rapid deficit correction crucial therapeutic steps to restore euvolemic hydration and renal perfusion in HHS [43]. Fluid losses usually range between 110 and 220 mL/kg, equivalent to 3–6 L in adults, and can reach 4000 mL/m^2^/day. The goal is to achieve full deficit replacement within 24 h, typically by administering normal saline (NS) at an initial rate of 20 mL/kg and repeating as needed, with a minimum of 40 mL/kg in the first 6 h until peripheral perfusion is restored. Transitioning to hypotonic 0.45 NS is reserved for fully resuscitated patients with persistent hypernatremia and hyperosmolality to avoid overcorrection and promote a gradual decrease, reducing the potential risk of brain edema [27]. Some authors recommend replacing urinary losses with an isotonic solution if the patient is hemodynamically unstable [21,38,40].

In addition to fluid management, monitoring and managing electrolyte imbalances, such as potassium and phosphate, are essential for a comprehensive treatment approach. Insulin therapy should also be initiated cautiously, as HHS patients may be more sensitive to insulin than those with DKA [40,43]. Moreover, addressing the underlying cause of HHS, whether it be infection, medication noncompliance, or other factors, is crucial for the prevention of future occurrences and long-term management.

Hemodialysis has been advocated by some for rhabdomyolysis and renal failure in patients who do not achieve the desired rate of serum Na and osmolality decrease. The gradual decline with dialysis can reduce complications [44]. Furthermore, close monitoring of the patient’s clinical and biochemical parameters is vital during the treatment process to ensure optimal outcomes and minimize potential adverse events.

Contrary to insulin therapy in DKA, exogenous insulin for HHS should be delayed and decreased, as there is usually no significant ketosis [45]. Fluid resuscitation alone substantially decreases serum glucose in HHS by intravascular volume expansion, producing a hemodilutional effect, tissue circulation improvement, increased metabolic and glucose utilization, and, finally, renal perfusion enhancement. The latter increases glycosuria; therefore, any additional and simultaneous insulin hypoglycemic effect could be detrimental by accelerating the glucose fall curve precipitously and exceeding the optimal desired serum glucose decline rate of 75–100 mg/dL/h (4.1–5.5 mmol/h) [46].

The overlaps and defining factors of the HHS and DKA differences approach can be emphasized through comprehensive and distinguished management steps included in a summarizing DI-FF-ER-EN-CE-S mnemonic, which serves as an invaluable guide for healthcare professionals to remember the specific management strategies unique to HHS. Furthermore, it underscores the importance of education and training for healthcare professionals in recognizing and managing HHS, including mixed cases with DKA, as a key factor in improving patient care. The mnemonic breaks down as follows:-‘DI’—Delay and Decrease Insulin Administration: HHS patients require a later initiation and a lower dose of insulin compared to those with DKA. This stage emphasizes the need to manage the decline in serum glucose, primarily through fluid resuscitation, before insulin therapy is initiated.-‘FF’— Fluid replacement emphasis: Fluid management is pivotal in HHS due to severe dehydration, which is often double that observed in DKA.-‘ER’—Electrolyte replacement, especially potassium, phosphate, and magnesium, are more pronounced in HHS and necessitate frequent monitoring and replacement, initially as often as every 2 h.-‘EN’—Encephalopathy: HHS can potentially lead to encephalopathy, underlining the need for controlled serum osmolality reduction. The regular monitoring of clinical and biochemical parameters, including close hourly glucose monitoring, is essential during treatment.-‘CE’—Cerebral Edema: While rare in HHS, cerebral edema requires prompt and aggressive therapy when it occurs.-‘S’—Systemic Multiorgan Failure: HHS can potentially lead to systemic multiorgan failure. For ease of awareness and screening, we categorize the associated risks into three mnemonic groups: the 3Rs (renal failure, respiratory distress, rhabdomyolysis), the 3Hs (heart failure, hypercoagulation, hyperthermia), and AP (arrhythmias, pancreatitis). Early recognition and management of those complications are crucial for improving patient outcomes.-Premature insulin use in under-resuscitated HHS patients with vasospasm may cause hypotension due to a prominent extravascular fluid shift. A rapid decline in the serum glucose of >100 mg/dL may occur in the first hours of fluid expansion alone [17]. Therefore, insulin infusion should be held temporarily if it was already initiated. Some authors advise adding low glucose (2.5–5%) if a significant decrease continues [47]. Once the serum glucose begins to fall at a rate below 50 mg/dL/h (2.7 mmol/L/h), an insulin drip is added to fluid therapy at a lower dose of 0.05–0.025 unit/kg/h. This approach contrasts with that used for patients with severe ketoacidosis typical of DKA, where insulin infusion is initiated earlier and at a higher dose (around 0.1 unit/kg/h) [48]. The initial goal of hyperglycemia level control for HHS is between 200 and 300 mg/dL, with an IV insulin drip titrated at a lower rate than in DKA. In comparison, the blood glucose goal for DKA on IV Insulin is 150–200 mg/dL. A hypernatremic sodium decrease of 0.5 meq/L/h is advocated in some studies [49]. Generally, electrolyte losses in HHS, including losses in potassium, phosphate, and magnesium, are higher than in DKA. Potassium losses are extreme despite the possible initial hyperkalemia secondary to acidosis ion shift [50]. If the serum potassium is <5 mEq/L with adequate renal function, it is recommended that patients are provided with 40 mEq/L of potassium phosphate/KCl. Magnesium may be needed for hypomagnesemia or hypocalcemia [40,43,45,47,48].

Dantrolene is the drug of choice for malignant hyperthermia in HHS, to stabilize myoplasm calcium [51]. Heparin prophylaxis to prevent deep vein thrombosis (DVT) is recommended in HHS patients who require a central line and are immobilized for 1–2 days [52].

Brain edema treatment should be initiated with a clinical diagnosis and should not require head CT confirmation since false negative results have been reported. Standard treatment includes head elevation, respiratory support, and specific therapy of Mannitol as the first line treatment [2,24,26]. Hypertonic saline is reserved for unresponsive cases after infusing 1 gm/kg of Mannitol. Bicarbonate administration should be avoided unless the pH is less than 6.9, which is not characteristic of HHS but can occur in severe DKA [53].

The resolution of HHS is achieved by rehydration, serum osmolality normalization, the restoration of a normal mental status in uncomplicated cases, and the stabilization of electrolyte derangements. If present, any underlying hyperglycemic precipitating illness, including infection, should be targeted early.

## 6. Prognosis

Worse HHS outcomes have been reported in comatose, hypotensive, and obese patients [54]. The level of serum osmolality has not been proven to have a predictive mortality value [55]. While cerebral edema accounts for most of the 0.15–0.35% mortality rate in DKA reported by some studies, HHS mortality is markedly higher, exceeding 10% in some reports, representing double the adult rate [56]. Common causes of high HHS mortality include severe dehydration, unreversed shock within 24 h of presentation, inadequate fluid resuscitation (<40 mL/kg in the first 6 h of treatment), electrolyte disturbances, hypertonicity, and multiorgan failure [57]. Grave complications predictive of death have been noted in unconscious and seizing patients and include rhabdomyolysis, malignant hyperthermia, renal failure, pancreatitis, persistent hypernatremia, and multiorgan failure [48,49,50,51]. In a limited retrospective study, delayed HHS diagnosis increased mortality to 60% [58].

## 7. Mixed HHS and DKA

The diagnostic landscape of diabetes-related conditions, particularly HHS and DKA, is intricate and complex. Traditionally, these two conditions have been discerned based on their distinctive classical differential characteristics, providing clear diagnostic pathways. However, in practical clinical scenarios, particularly in pediatric cases, this differentiation is not always so clear-cut. Increasingly, unique and challenging cases have emerged that present with a blend of symptoms or biochemical criteria traditionally associated with either HHS or DKA. This creates a unique spectrum of conditions that are not wholly HHS or DKA, but rather a coalescence of both. This has led to the identification of hybrid conditions, such as hyperosmolar DKA and ketotic HHS. These cases embody the cumulative risks of both HHS and DKA, highlighting the clinical complexity and potential severity of these overlapping conditions [5,59]. In certain manifestations of HHS, we may observe a phenomenon of mild to moderate acidosis, even in the absence of significant ketosis. Such occurrences can be perplexing given the conventional understanding of HHS. The explanation for this atypical presentation can often be traced back to physiological processes, like hypoperfusion and anaerobic metabolism. In such scenarios, due to an insufficient oxygen supply or impaired blood flow, the body switches to anaerobic metabolism, resulting in the production and accumulation of lactic acid and, thereby, lactic acidemia [58,59]. Furthermore, a noteworthy clinical observation is the exacerbation of hyperglycemia in patients with DKA who attempt to alleviate their extreme thirst. Thirst is a common symptom of DKA, driving patients to consume beverages that are often high in carbohydrates. Paradoxically, this well-intentioned attempt to mitigate their symptoms can lead to severe hyperglycemia, further complicating their condition [60].

Some HHS cases present along a continuum between two hyperglycemic poles, dependent at least on the degree of insulin deficiency or its tissue sensitivity, dehydration severity and fluid intake, symptom duration, and patient status [61,62]. These factors affect the extent of ketosis and acidosis or range of serum glucose elevations, dehydration, hyperosmolality, and DKA. Patients presenting with some combination of HHS and DKA are particularly challenging, and diagnosis and treatment require a higher degree of vigilance with attention to the risks and treatment priorities of the HHS and DKA components.

The management of these patients should be guided by the biochemical profile and clinical judgment, evidence from the management of isolated HHS and DKA, and a strong focus on individualized patient care. Based on the predominant findings, in addition to the recognized deficits and risks, the treatment of patients with mild DKA and significant hyperosmolar dehydration should begin with saline boluses to restore fluid losses, with the simultaneous monitoring of the rehydration volume response, electrolyte derangement precautions, and supplementation. Insulin therapy should be delayed until there is no significant decrease in hyperglycemia as in classical HHS, starting at half the typical 0.1 unit/kg/h insulin infusion dose used in DKA, with glucose and saline fluid adjustments based on the hyperglycemia and electrolyte status.

In the context of these complex overlapping cases, early and frequent interdisciplinary collaboration involving endocrinology, nephrology, and critical care specialists can optimize patient management and outcomes. In challenging overlapping cases of HHS and severe DKA, we propose dynamic adjustable combined therapy that addresses both conditions based on which state is predominant, with particular attention to HHS complications, coupled with close clinical and laboratory monitoring, starting with fluid boluses at volumes and repeats determined by the severity of the HHS and DKA and their manifestations. The insulin dose should be lower than that used for DKA and decreased to 0.05–0.1 unit/kg/h. The risk of cerebral edema is higher in mixed cases than in isolated HHS, warranting greater awareness and clinical monitoring. Moreover, ongoing patient and family education about self-management strategies, including the proper monitoring of blood glucose levels and the timely administration of insulin, can play a significant role in preventing future episodes of HHS or DKA. To aid the understanding and guide the clinical management of these convoluted diabetic states, Figure 3 offers a comprehensive summary. This figure outlines the fundamental defining criteria of the diabetic states discussed and aligns them with the corresponding initial therapeutic interventions. This synthesis serves as an invaluable tool in navigating these multifaceted conditions, allowing for an effective and targeted therapeutic approach.

## 8. Conclusions

In conclusion, the differentiation of HHS from DKA is vital for proper management and improved patient outcomes. The growing prevalence of HHS in diverse populations, including children and type 1 diabetics, emphasizes the need for awareness and an accurate diagnosis. The unique monitoring and treatment measures required for HHS, distinct from DKA protocols, necessitate the creation of standardized treatment regimens and guidelines. By focusing on a prompt diagnosis, understanding predictive prognostic values, and implementing early, targeted treatment measures, such as utilizing the DI-FF-ER-EN-CE-S mnemonic with the 3Rs, 3Hs, and AP complications acronyms monitoring, clinicians can navigate the complexities of these hyperglycemic emergencies, ultimately reducing risks and enhancing patient care.

## Figures and Tables

**Figure 1 children-10-01773-f001:**
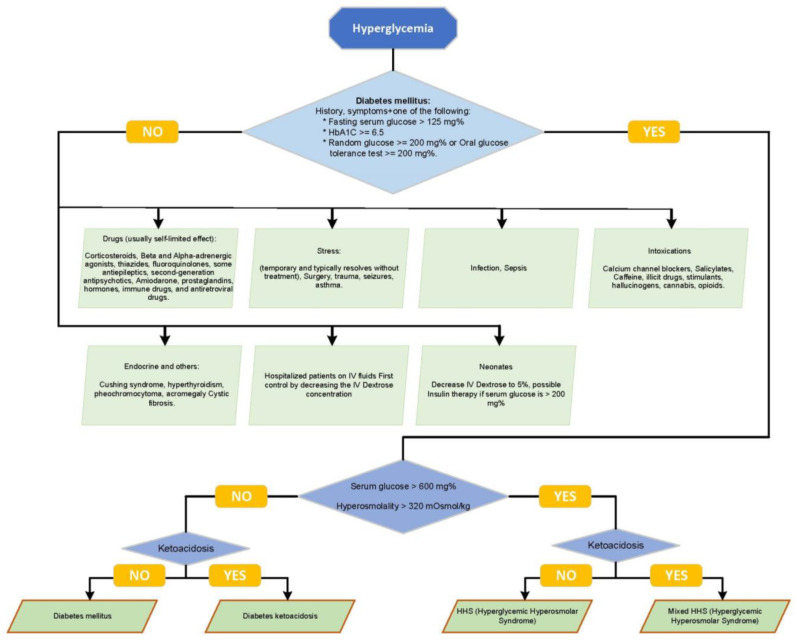
Algorithmic classification illustrating the various etiologies and hyperglycemic states, including a detailed differentiation between HHS and DKA, to guide clinicians in the evaluation and management of these complex conditions.

**Figure 2 children-10-01773-f002:**
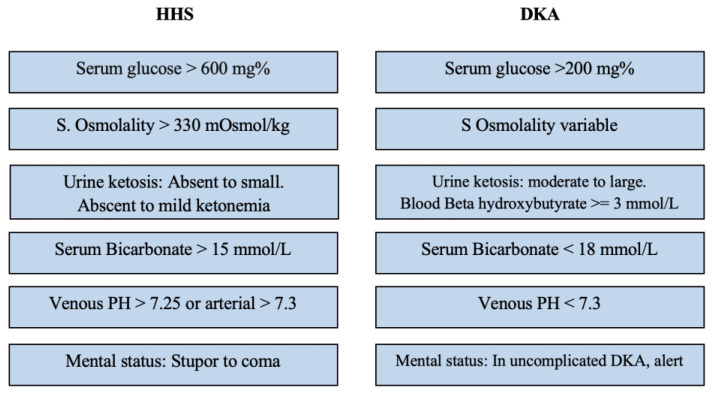
Comparative visualization of the distinct characteristics and diagnostic criteria between HHS and DKA [20,21,22].

**Figure 3 children-10-01773-f003:**
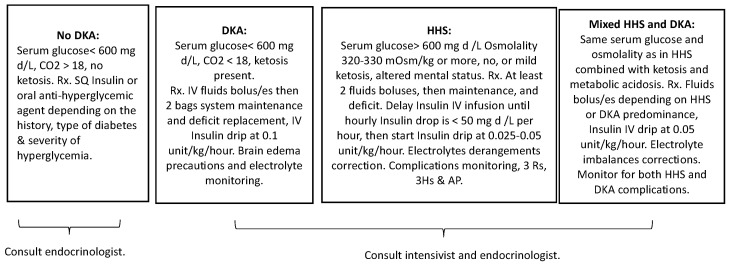
A comprehensive summary outlining the fundamental defining criteria of HHS and DKA, aligned with the corresponding initial therapeutic interventions, serving as a tool for an effective and targeted therapeutic approach in the clinical management of these conditions. Abbreviation: Rx, Prescribe. SQ, Subcutaneous.

**Table 1 children-10-01773-t001:** Pediatric Endocrine Society HHS Diagnostic Criteria.

Plasma glucose exceeding 600 mg/dL (33.3 mmol/L) Serum osmolality > 330 mOsmol/kg, serum bicarbonate > 15 mmol/L, arterial pH > 7.3 or venous pH > 7.25 Absence or low serum ketones (on dipstick < 15 mg/mL or 1.5 mmol/L), which could partially represent starvation ketosis Altered mental status

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
