# Peer review of "Pediatric Hyperglycemic Hyperosmolar Syndrome: A Comprehensive Approach to Diagnosis, Management, and Complications Utilizing Novel Summarizing Acronyms"

_children, 2023, doi:10.3390/children10111773_

Round 1
Reviewer 1 Report
This paper gives a comprehensive review and knowledge on HHS and its common shared characteristics with DKA. The diagnosis, complications, and management are well introduced. The paper provides a good insight into the awareness and screening of HHS. However, major revisions are needed before consideration for publication.
Comments:
1. Introduction: More background information on HHS and DKA would be appreciated. What's the prevalence? What is the current state of research? Why is it important to increase awareness and facilitate screening of HHS?
2. All figures are missing legends. Also need to label the abbreviations in the legends.
3. Line 77-78: cite these literatures.
4. For contents in Fig 2: Are these established diagnosis criteria? Any references?
5. Line 154-159: Here the authors spend a lot of words explaining the diagnosis criteria of brain edema. Why is it important? If it rarely occurs in HHS, can these signs also be considered to distinguish between HHS and DKA?
6. For DKA and HHS, no need to introduce the full terms every time.
7. From line 162 the authors started to discuss cerebral edema again while in the above section brain edema was already discussed. Lines 162-164 seem a little repetitive. Also, if this is an important complication why don't the authors discuss its diagnosis criteria here instead of the previous section?
8. In the conclusion, lines 393-397 just repeat the intro. Here a brief summary is needed instead of repetitive information. The conclusion is to summarize the key take-aways.
The overall English quality is acceptable.
Author Response
1. Introduction: More background information on HHS and DKA would be appreciated. What's the prevalence? What is the current state of research? Why is it important to increase awareness and facilitate screening of HHS?
Intro addended to reflect this feedback
2. All figures are missing legends. Also need to label the abbreviations in the legends.
Legends added
3. Line 77-78: cite these literatures.
Lines cited
4. For contents in Fig 2: Are these established diagnosis criteria? Any references?
References were provided in surrounding text but also added to figure legend for convenience.
5. Line 154-159: Here the authors spend a lot of words explaining the diagnosis criteria of brain edema. Why is it important? If it rarely occurs in HHS, can these signs also be considered to distinguish between HHS and DKA?
Paragraph addended to in response to this feedback
6. For DKA and HHS, no need to introduce the full terms every time.
Full terms removed after introduced earlier in the paper.
7. From line 162 the authors started to discuss cerebral edema again while in the above section brain edema was already discussed. Lines 162-164 seem a little repetitive. Also, if this is an important complication why don't the authors discuss its diagnosis criteria here instead of the previous section?
Paragraph addended to in response to this feedback
8. In the conclusion, lines 393-397 just repeat the intro. Here a brief summary is needed instead of repetitive information. The conclusion is to summarize the key take-aways.
Conclusion addended in response to this feedback

Reviewer 2 Report
I would like to thank Zahran et al. for their essay on pediatric HHS. Even if their acronyms could be of some use to the practicing pediatrician, I think that the paper has two major drawbacks. First, it looks nothing like a medical journal paper but rather a review chapter in a book. In fact, it seems like the authors came up with the acronyms and wrote a review on HHS around these acronyms. Second, I really don’t find these acronyms to be very helpful. For example, EN and CE correspond basically to the same possible HHS complication which, in fact, occurs much more rarely than in DKA. In addition, I have some minor suggestions for the improvement of the manuscript:
· All figures need to have explanatory text
· Figure 1 needs to be adjusted in order to be readable. The font-size is too small for someone to read
· Lines 38-40: please rephrase the sentence
· Lines 59-60: please rephrase the sentence
· Lines 70-71: DKA is not “commonly” associated with these features, they are its diagnostic criteria
· Lines 96-100: This sentence repeats what is analyzed in the previous two sentences, ie. the pathomechanism of ketoacidosis absence in HHS. I would suggest you merge the information in order to avoid repetition
· Line 125: DKA can develop in hours or a couple of days but usually has a more protracted course over several days or weeks. I suggest you rephrase the sentence
· Lines 129-130: Usually, such signs and symptoms appear in more severe cases of DKA and is not the usual presentation
· Line 133: “In some research”: please rephrase
· Lines 150-160: This referral to the paper by Muir et al. is about cerebral edema, which is described at the next paragraph. I suggest you merge this info with the ones given under the heading “4. Complications in HHS and DKA”
· Line 154: “altered mental status”
· Line 156: Put the reference in brackets, the same in line 191
· Lines 269-276: Encephalopathy and cerebral edema refer to the same possible complication of HHS, which in fact is much rarer than in DKA
· Line 307: “Bicarbonate administration” instead of “bicarb”
· Line 327-328: Abbreviations already described in the text, the same in lines 393-394
· Line 331: “is not” instead of “isn’t”
Minor suggestions, well written.
Author Response
I would like to thank Zahran et al. for their essay on pediatric HHS. Even if their acronyms could be of some use to the practicing pediatrician, I think that the paper has two major drawbacks. First, it looks nothing like a medical journal paper but rather a review chapter in a book. In fact, it seems like the authors came up with the acronyms and wrote a review on HHS around these acronyms. Second, I really don’t find these acronyms to be very helpful. For example, EN and CE correspond basically to the same possible HHS complication which, in fact, occurs much more rarely than in DKA. In addition, I have some minor suggestions for the improvement of the manuscript:
1.All figures need to have explanatory text
Figures and Table updated
2.Figure 1 needs to be adjusted in order to be readable. The font-size is too small for someone to read
Figure 1 was enlarged for readability
3.Lines 38-40: please rephrase the sentence
Entire section rewritten as per alternate reviewer feedback as well
4.Lines 59-60: please rephrase the sentence
Entire section rewritten as per alternate reviewer feedback as well
5.Lines 70-71: DKA is not “commonly” associated with these features, they are its diagnostic criteria
Wording updated
6.Lines 96-100: This sentence repeats what is analyzed in the previous two sentences, ie. the pathomechanism of ketoacidosis absence in HHS. I would suggest you merge the information in order to avoid repetition
Section updated and feedback incorporated
7.Line 125: DKA can develop in hours or a couple of days but usually has a more protracted course over several days or weeks. I suggest you rephrase the sentence
Sentence rephrased
8.Lines 129-130: Usually, such signs and symptoms appear in more severe cases of DKA and is not the usual presentation
Wording updated
9.Line 133: “In some research”: please rephrase
Wording updated
10 Lines 150-160: This referral to the paper by Muir et al. is about cerebral edema, which is described at the next paragraph. I suggest you merge this info with the ones given under the heading “4. Complications in HHS and DKA”
Information merged
11.Line 154: “altered mental status”
Wording changed
- Line 156: Put the reference in brackets, the same in line 191
Formatting updated
- Lines 269-276: Encephalopathy and cerebral edema refer to the same possible complication of HHS, which in fact is much rarer than in DKA
Section updated
14.Line 307: “Bicarbonate administration” instead of “bicarb”
Wording updated
15.Line 327-328: Abbreviations already described in the text, the same in lines 393-394
Wording updated
16.Line 331: “is not” instead of “isn’t”
Wording updated

Round 2
Reviewer 1 Report
The authors have addressed some of the issues raised earlier. However, there are still minor issues to be addressed.
1. Full term of DKA still occurs after first time introducing. Check carefully.
2. To me the discussion of cerebral edema is still repetitive and seems everywhere.
When editing the original manuscript, the authors need to highlight the parts edited or change a color to make it clear to reviewers.
Author Response
Combed through the areas where DKA is spelled out and fixed.
Tried to make the discussion on cerebral edema less convoluted

Reviewer 2 Report
Still, a book chapter rather than a journal paper. The authors made no change whatsoever regarding this specific issue. In addition, still, a rather unhelpful acronym.
Author Response
Updated discussion regarding cerebral edema and made some editing changes to make it seem less like an editorial and more of a review on the subject
